# Rectal swabs vs bulk faeces PCR testing for the diagnosis of enteric conditions (RecSwabFaeces): Protocol for a single group diagnostic accuracy comparative trial

Madisen Sadie Roser[1], Megan K. Young[2,3,4], Dustylee Williams[1], Shradha Subedi[5], Donna Barnekow[1], Alyssa Pyke[6], Jacob Tickner[6], Glen Hewitson[6], Michael Thomas [6], Gulam Khandaker[7], Sanmarie Schlebusch[6], Nicolas Roydon Smoll [1]*

1 Sunshine Coast Public Health Unit, Maroochydore, Queensland, Australia, 2 Metro North Public Health Unit, Brisbane, Queensland, Australia, 3 School of Medicine and Dentistry, Griffith University, Southport, Queensland, Australia, 4 School of Public Health, University of Queensland, Brisbane, Queensland, Australia, 5 Sunshine Coast University Hospital, Birtinya, Queensland, Australia, 6 Pathology Queensland, Brisbane, Queensland, Australia, 7 Central Queensland Hospital and Health Services, Rockhampton, Queensland, Australia

* nicolas.smoll@health.qld.gov.au

## Abstract

Obtaining a stool specimen for diagnostic testing of enteric conditions (e.g., gastroenteritis) can be a challenging and unpleasant experience. A person is required to obtain a sample pot from a healthcare location, return home and wait until they have a bowel motion, and then deal with the challenges of returning the sample to the clinic or pathology centre. This trial aims to identify whether the simpler approach of obtaining a rectal swab is effective for diagnosing enteric conditions. Recruitment will take place in a variety of settings, including suspected norovirus clusters, Hepatitis A clusters, in both hospital, and community settings. We will compare paired stool and rectal swab sample polymerase chain reaction (PCR) testing to determine whether rectal swabs are a reliable proxy for faecal sampling. Persons who would normally be provided with a faecal specimen container, will also be provided with a rectal swab for self-collection or clinician-collection (within 24 hours of the bulk faeces collection). We will assess the sensitivity, specificity, positive predictive value, negative predictive value using standard confusion matrix. The gold standard reference will be considered the bulk faeces PCR tests. We hypothesise that rectal swab PCR testing will be equally as effective as bulk faeces PCR testing. If successful, rectal swab PCR testing could be implemented as routine practice with several key benefits. Firstly, it would improve the patient experience by conveniently enabling stool collection at the point of care, which is rarely possible currently. Secondly, it would reduce healthcare costs, and streamline collection processes by eliminating the need for persons to return to a clinic to deliver specimen. Thirdly, the ease of testing will likely increase

**Data availability statement:** No datasets have been generated or analysed as this manuscript is a protocol. All relevant aggregated data produced from this study will be made available upon trial completion through future publications. Data that has not been deidentified will never be shared publicly in alignment with ethics requirements.

**Funding:** The author(s) received no specific funding for this work.

**Competing interests:** There are no competing interests to be declared.

**Abbreviations:** AE, Adverse event/s; ANZCTR, Australian New Zealand Clinical Trials Registry; GP, General practitioner; HREC, Human Research Ethics Committee; PI, Principal Investigator; QA, Quality assurance; SAE, Serious adverse event; SAR, Serious adverse reaction; TSC, Trial Steering Committee.

testing rates and compliance, leading to better diagnoses and more accurate clinical care. The protocol described has the potential to revolutionise daily clinical practice.

**Trial registration**: Australian New Zealand Clinical Trials Registry (ANZCTR): ACTRN12624000095561.

## Introduction

Testing persons with gastroenteritis-like illnesses necessitates the provision of a faecal sample. This method is not without drawbacks; in particular, the very nature of defecation means that samples cannot be provided 'on demand' in a physician's office or to a research nurse and are reliant on appropriate collection by patients themselves [1]. Patients in hospitals with diarrhoea commonly use the toilet before healthcare workers can take a stool sample. Faeces sampling may also present additional complexities, including the logistical challenges of having to transport samples between the patient's home, the clinic and the laboratory, and considerable patient reluctance to providing samples. Rectal swabbing has been shown to be a valid alternative in several research and clinical settings [2,3].

Rectal swabbing is standard practice in Australia and overseas for the surveillance and screening of multi-resistant organisms. National U.K and Australian guidance mandates the use of rectal swabs for screening for intestinal colonisation with carbapenemase-producing Enterobacteriaceae for at-risk patients admitted to healthcare settings [3,4]. The key benefits of rectal swabs include the ease with which they can be administered and transported, and the high levels of acceptability to patients/research study participants [3].

Self-collection as a process has been shown to be an accurate testing methodology in other settings. Human papilloma virus self-collection has been shown to be just as accurate, or even more sensitive than clinician collected samples [5–7]. The accuracy of rectal swabs vs bulk faeces testing has been tested extensively using culture mediums, but although some research has been done in this area since the advent of polymerase chain reaction (PCR) testing methodology, routine use has not been adopted.

Studies have long compared rectal swabs vs bulk faeces testing. The results are varied, depending on the year the study was performed. An early study examining 700 specimens tested with MacConkey's medium in 1940 during a Sonne dysentery outbreak (*Shigella sonnei* sp. outbreak) found that rectal swabbing was far superior to bulk faeces culture to detect an organism [8]. Rectal swabbing became the standard for a period of time This was disputed later in 1954 after a group examined 3,183 index cases looking for a variety of organisms to which they concluded that the *"…626 [bulk faeces] specimens all showed abnormalities which could be demonstrated only by microscopy or simple chemical methods, for which the rectal swab is quite unsuitable"* [9]. Since then, huge changes have occurred to our detection methods (e.g., culture media) and the advent of PCR testing for multiple organisms at once.

More recent studies demonstrate that the differences between rectal swabs and faecal samples is driven by the type of test (bacterial microscopy and culture vs viral PCR). A phase I clinical study for a live oral attenuated *Salmonella* Typhi vaccine strain found that when compared with faecal culture, rectal swabs were 64% sensitive and 90% specific [10]. A recent study on paired samples found that for bacteria, the sensitivity was 86.5% (95% CI 79.5%, 91.8%) when PCR was performed and 61.4% (95% CI 52.4%, 69.9%) when culture for bacteria was performed, indicating better performance for PCR [11].

It has been shown in small studies that rectal swabs are likely accurate. Studies have now shown that during testing for various enteric organisms (viruses and bacteria) in Rwandan children there was no significant difference in detection rate between faeces and rectal swabs for any agent, reflecting that pathogen concentration was far above the limit of detection in the majority of cases [12]. A similar paired sample study focusing on norovirus found a sensitivity of rectal swabs that was greater than 97%, and in one case detected norovirus in the swab, but not in the bulk faeces [13].

The outcome of this project will transform clinical practice for doctors and patients. It will make testing far more convenient and allow for point-of-care testing to become a reality for enteric organisms. Currently, many people leave the emergency department without a faecal test because they could not produce a specimen while they were with the clinician. Rectal swabbing would make the diagnosis, and better care available for these patients. Public Health practitioners would be able to perform mass testing after small or large scale outbreaks, such as those caused by hepatitis A, norovirus or other organisms found in faeces [14,15]. Doctor's in the emergency department would be able to obtain swabs on demand for testing that would expedite time spent in ED and, with the earlier diagnosis, length of stay in hospital. Most importantly, for patients, it means that they no longer must go home, wait until they need to have a bowel motion, and then return to the hospital or GP clinic with their sample, as it can be done on demand, on-site (in a private situation). Thus, rectal swabbing can transform clinical practice through earlier diagnosis, shorter length of stay's and greatly improving the patient experience. This study aims to demonstrate that self-collected or clinician collected rectal swabs are a valid alternative method of testing when compared to bulk faeces testing for a wide range of enteric organisms such as bacterial, viral, and parasitic/protozoan. The intended outcome of this project is to replace bulk faeces testing with rectal swabbing in daily clinical and public health practice.

## Methods

This is a single group paired sampling prospective trial comparing the diagnostic accuracy between an established method and an alternative method of diagnosis. Participants of the study are required to use both sampling methods to allow for comparison of results. A broad timeline of the trial is presented in Fig 1. This trial was registered with the Australian New Zealand Clinical Trials Registry (ANZCTR) (registration number: ACTRN12624000095561) on 02/02/2024. Ethics approval was granted for this trial by the Metro North Human Research Ethics Committee (HREC) (EC00168) on 19/12/2023. Written or verbal consent to participate will be obtained from each participant. Records of consent will be stored securely.

### Study setting

The study aims to recruit patients in the community (primary care or public health responses) and in public hospitals. The mode of collection will either be a self-collect rectal swab (done by the patient themselves) and bulk faeces sample or clinician collect rectal swab and bulk faeces sample within a clinical environment. The generalisability of this trial is supported by its relatively broad inclusion and exclusion criteria, which are designed to reflect routine clinical practice. Because the trial aims to evaluate the use of rectal swabs as an alternative to bulk faeces sampling, eligibility is not restricted (e.g., based on symptom presentation), allowing for the inclusion of participants with diverse clinical profiles. This inclusive approach will mimic routine clinical practice, and facilitate recruitment across a broad range of demographic, clinical, and socioeconomic backgrounds, enhancing the external validity of the findings. Vulnerable groups, such as

| | Study period | | | | |
|---|---|---|---|---|---|
| | Enrolment | | | Post-collection | Trial close-out |
| TIMEPOINT | 1-12 months | 12-24 months | 24+ months | ~30-36 months | 36 months |
| Eligibility screen | X | X | X | | |
| Informed consent | X | X | X | | |
| Send sample collection packs | X | X | X | | |
| Sample collection | X | X | X | | |
| Samples collected | X | X | X | | |
| Samples submitted | X | X | X | | |
| Laboratory analysis | X | X | X | | |
| Follow-up phone call (occurs after individual sample submission) | X | X | X | | |
| Interim analyses | At 12 months | At 24 months | | | |
| Recruitment completed | | | | X | |
| Final data analysis | | | | X | |
| Manuscript writeup | | | | | X |
| Publication and dissemination | | | | | X |

**Fig 1. SPIRIT schedule.** SPIRIT schedule of enrolment, interventions, and assessments.

comorbid or immunocompromised individuals, are identified in the recruitment phase and this data is recorded to be used in the analyses. The lower age limit of 18 years has been set due to ethical and regulatory considerations in recruiting children and performing rectal swabs on a vulnerable group. While children are not included in this study, the trial findings relevant to rectal swab accuracy are expected to be applicable to all ages.

Inclusion criteria:

• Persons aged 18 and older.

• Able to consent to and perform the additional rectal swab test.

• Close contact or person exposed, suspected, or confirmed case associated with an outbreak of an enteric organism. E.g., hepatitis A outbreak, foodborne outbreak.

• Chronic carrier of an enteric organism that requires testing.

• Person involved in a clinical encounter where they are or are likely suffering from a disease caused by an enteric organism. (E.g., person presenting to the GP or Emergency Department with diarrhoea or jaundice)

Exclusion criteria:

- Persons unlikely (as judged by the clinician) to be harbouring an enteric organism.

- Persons where collection of a rectal swab is medically contra-indicated.

## Consent

In all study settings, the potential participant will be asked by the treating clinician for consent to be contacted by the local study coordinator. After which the local study coordinator/study nurse will contact the person to obtain consent for inclusion into the study.

There will be two ways that a participant can consent to participating (verbal, written). Verbal consent will be obtained over the phone with a study coordinator or study nurse, as is commonly done in outbreak situations. The study nurse will read the patient information sheet to the participant over the phone, ensure the participant understands the process and document this in a database held on Queensland Health data servers. The testing pack will be sent to the participant and will include the patient information sheet about the study and all contact details for questions about the study. Furthermore, implicit consent will occur when the participant obtains both samples (the participant can decline to participate by not providing the rectal swab) and returns one or both samples to the pathology laboratory.

If the testing is done in person (a study clinician is present), verbal, written and implicit consent (the patient will obtain the sample themselves and return it to the study coordinator) will be obtained using the patient information sheet, and the signature document will be kept, with a copy offered to the participant.

Opting out of the study will be possible for participants at all stages of the study, up until the analysis of the research. For example, a participant may agree to participation over the phone, then, once they receive the full pack with the Patient Information Sheet, and they read over it in detail, they can withdraw from the study by calling us and withdrawing, or, not sending the rectal swab, and then withdrawing after we check in with them after the rectal swab was not returned.

## Risks and harm

Risks associated with the rectal swab sample collection process include discomfort or pain, and in rare cases, infection. In some cases, participants may experience minimal bleeding. There is a low to medium risk of these events occurring if sample collection is conducted properly. These risks are outlined in the patient information and consent form, ensuring that participants are fully informed prior to their involvement. To mitigate these risks, strict adherence to hygiene and sterile techniques is emphasized during both self-collection and clinician-guided collection.

## Trial process

The site study coordinator will provide participants with oral and written information about the procedure, including its purpose, what to expect (2 tests being performed), and potential risks and then obtain verbal or written informed consent. The participant will ensure all necessary materials are present in the pre-prepared packs which will include sterile rectal swabs, bulk faeces collection sample container, gloves, pre-filled pathology form and labeling supplies. The participant or clinician will conduct the procedure in a private, clean, and comfortable setting to maintain participant dignity and comfort (preferably a toilet).

**Collection process.** Participants are encouraged to perform thorough hand hygiene before and after the procedure and wear the disposable gloves provided in the packs to maintain sterility and prevent contamination. Ideally the participant will swab first and then provide a bulk faeces sample second. The swab will be gently inserted into the rectum, approximately 2–3 cm (about 1 inch). The swab should be rotated for 10–15 seconds to ensure adequate sampling. The swab should be carefully withdrawn, ensuring not to touch any external surfaces to avoid contamination. The participant will defecate directly onto a paper or plastic liner or into the collection device. The participant will be informed not to

urinate at the same time to avoid contaminating the sample. Using the provided scoop or spatula, the participant will then transfer a portion of the faeces into the provided collection container. A sample size of approximately 5 grams (about the size of a walnut) should be collected. The containers (rectal swab and bulk faeces) should be securely sealed to prevent leakage. Containers will be properly labelled with the participant's unique identifier, date, and time of collection.

**Control sample (bulk faeces).** The bulk faeces sample is used as the control sample in this study as this sample type is currently used in routine clinical practice. In this context, bulk faeces PCR testing is considered the gold standard. It is acknowledged that bulk faces PCR testing may be associated with technical or interoperability limitations. To minimise the uncertainty that this could pose on testing results, paired samples that do not produce the same results will be managed internally according to laboratory protocol.

**Sample handling, transport and storage.** After the sample has been properly labelled, the sample will be transported to the laboratory as soon as possible to maintain sample integrity. The sample may be stored in a refrigerator until it can be sent to the lab and/or processed. A clear chain of custody for the sample will be maintained, documenting each step from collection to laboratory analysis.

**Follow-up.** Each person will receive a follow-up phone call after the test has been completed. The participant will be asked structured questions regarding the testing procedure:

- "Which sample did you collect first?" (bulk/swab). This will help to validate the timing of sample collection, ensuring that samples that were collected close together (e.g., the participant records the same collection time for the swab and container) in time can be ordered.

- "Did you experience an adverse event during or after collecting the swab sample? (yes/no). If so, what was the nature of the event?" (qualitative). The adverse event will then be classified according to classification definitions described in 'adverse event reporting and harms'.

- "Which test did you prefer?" (bulk/swab). "Why?" (qualitative)

### Outcomes

Primary Outcomes: Assess sensitivity, specificity, positive and negative predictive value, positive and negative agreement, accuracy, and receiver operating curves (at various CT value cutoffs), to determine the efficacy the new rectal swab approach.

Secondary outcomes: Acceptability of the rectal swabs in comparison to the gold standard of bulk faeces testing. This will be measured by asking participants which sampling method they preferred, and collecting information from participants in relation to any adverse events that may have occurred.

### Timeline

Participant recruitment is expected to commence in early 2025. Recruitment and data collection are expected to be completed within two years. Results are expected shortly after, placing the trial completion date in late 2027 or early 2028. It is expected that an average of 3–6 swabs will be obtained per week across collection sites. Some situations, such as outbreaks or mass gathering events, will allow for greater numbers of swabs to be collected in a small timeframe. After participants return the test samples, they will receive a follow-up phone call. Results of the bulk faeces test will be returned to the patient by the clinician that requested the test.

### Sample size calculation

A sample size analysis with a wide range of input parameters, as described by Hajian-Tilaki 2014, demonstrates that less than 600 samples will be sufficient to cover a range of diseases (Fig 2) [16]. Assuming that 30% of samples will contain

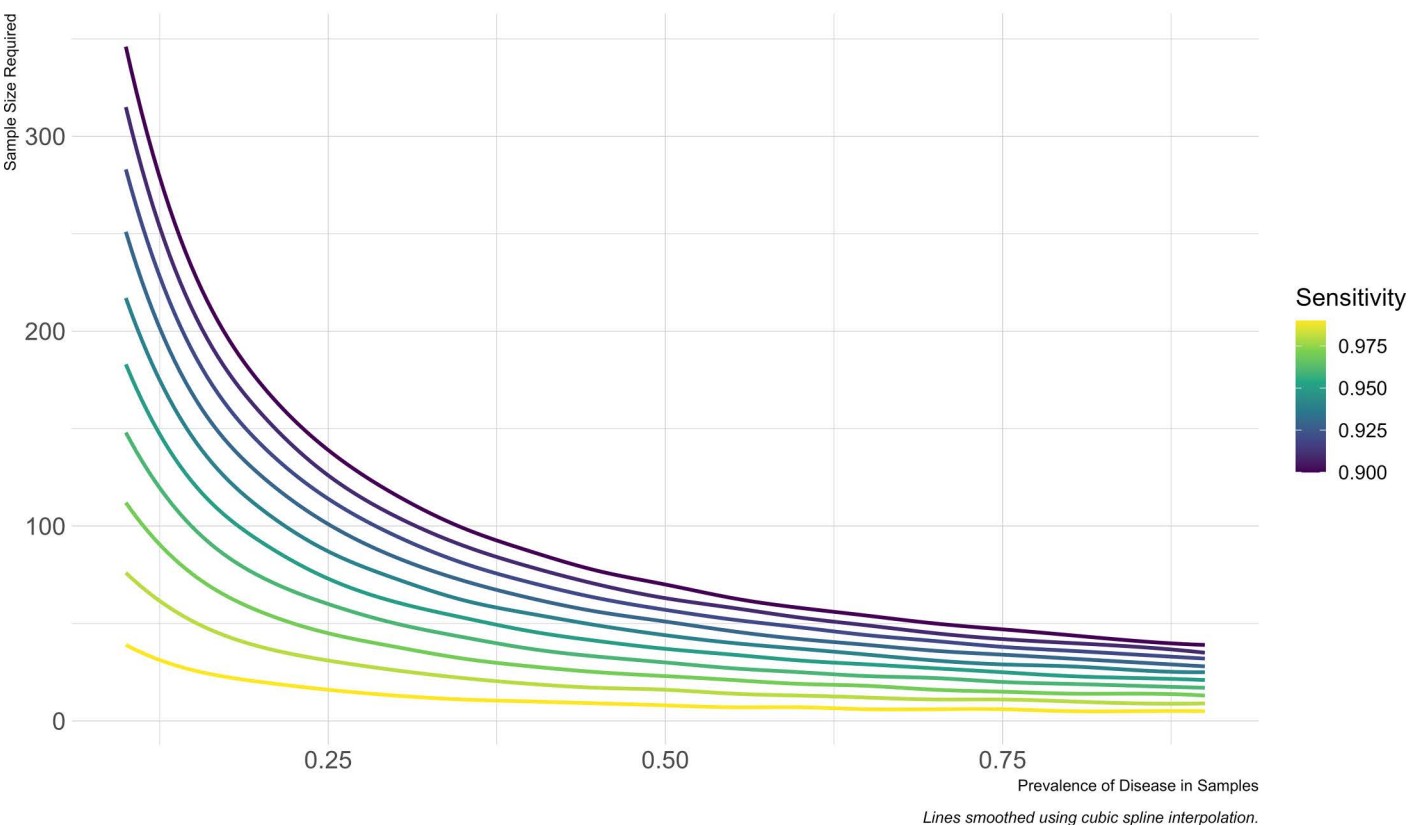

## Power Calculation - Diagnostic Sensitivity (Smoothed)

Sample size required to estimate sensitivity with 95% confidence and 10% absolute error

*Lines smoothed using cubic spline interpolation.*

**Fig 2. Sample size analysis.** Includes a range of parameters. The "Sensitivity priors" represents the approximate expected sensitivity of our method, and the "proportion positive" represents the proportion of samples that will contain an organism, which also means that should test positive.

an organism ("proportion positive") with an expected sensitivity of at least 95% ("sensitivity priors") means we will require a sample size of no more than 61 paired samples. As the prevalence of disease in samples decreases, the sample size required is higher. For a sensitivity of 95% and a prevalence of 20%, the sample size increases from 61 to 92. These estimates are based on general sensitivity assumptions and a simplified disease prevalence model, with the literature reporting non-specific prevalence to be approximately 25%, and sensitivity estimates typically >90% [17–20]. These estimates may change slightly as recruitment progresses and the distribution of pathogens becomes clearer. To address potential variability across different pathogens, stratified analyses will be conducted. Power calculations for key pathogen subgroups indicate that, for a sensitivity of 95% (with a range of 90–99%), a minimum of 73 positive samples is required per pathogen subgroup.

Constrained resources (e.g., funding), demands a higher pre-test probability at the time of testing to estimate with confidence. For example, if we only have funding for 40 persons tested, we will elect to test persons with clear known linkages to a person with the illness (symptomatic family member of someone diagnosed with norovirus), or we would perform clearance testing in persons with known disease using these methods (the guidelines call for clearance testing for a variety of illnesses). With greater funding, testing can approximate normal clinical encounters and have less restrictive criteria

for testing. If the funding is limited to 40 samples, we will endeavour to control the testing method, aiming for in-clinic self-testing.

## Recruitment

The recruitment strategy will be multifaceted to obtain samples from a range of representative clinical settings, e.g., outbreaks, community, and in-hospital. The strategy will involve a central study coordinator supporting acquisition of samples, coordinating the multiple sources. For outbreaks, the study coordinator will be able to send sample collection packs to the patient, family, and/or symptomatic members of an outbreak via express post. In community settings, the study coordinator can obtain samples from GP practices that are seeing patients who were recently inpatients in hospital – identified from admission lists, or our Infectious Diseases Physician study collaborators. Inpatients, especially those which represent persons with a high pre-test probability, will be identified, recruited via their treating practitioner, and then consented to take part in the study. Consent will be performed verbally over the phone by a study coordinator (records kept locally), and/or with a physical form sent in a return envelope.

## Laboratory testing

Faeces will be collected in standard collection containers and rectal swabs utilised will be dry cotton swabs. Samples will be registered and transported chilled using existing referral pathways. Biological materials will be stored in the laboratory in accordance with NPAAC minimum requirements and will be discarded once all required testing on these samples has been completed for assay validation, research and testing required for public health and/or clinical investigations. Faeces and rectal swabs samples will be subjected to nucleic acid extraction using the QIAGEN EZ or KingFisher FLEX instruments. Real-time RT-PCR will be performed to identify hepatitis A or E virus-positive specimens, using existing methods at the Public Health Virology laboratory, Queensland Health. Extracted RNA from hepatitis A or E virus RT-PCR positive samples will then be used to perform partial or whole genome sequencing. Paired rectal swabs and faeces samples from individual patients will be sequenced in parallel on the same sequencing run, using either Nanopore or Illumina sequencing chemistries and instruments. The partial or whole genome sequences generated from each patient will be analysed and firstly aligning with references sequences to determine genotype and then their nucleotide homology and phylogenetic relatedness will be compared to determine whether variation between the sequences from each respective sample type (i.e., faeces vs rectal swab) exists.

## Data collection and management

Laboratory diagnostics will provide the majority of the outcomes using established testing methods. Pathologists will provide the diagnosis for samples. We will be collecting information about demographics, symptomatology, setting it was collected in, type of collect (self-collect vs clinician collect), and two patient reported outcomes by firstly asking the patient if they had an adverse event during or after the testing, and the nature of the event if present. Secondly, we will ask the patient which test they preferred and follow-up with why. Data analysis will be conducted by members of the research team.

Participant-level dataset will not be publicly accessible. Statistical code and full protocol will be publicly accessible.

**Data storage, access, and de-identification.** This study will collect data related to routine clinical practice (bulk faeces testing), as well as research data (rectal swab results). The results of these tests will be kept in a clinical database (AUSLAB, within Pathology Queensland) as well as within local study databases (Excel spreadsheets or Redcap Databases) for operational and analytical purposes. At all times, data will be kept within Queensland Health databases and will not be shared with external entities. Only Queensland Health staff (includes Pathology Queensland) will have access to the clinical and research data.

Data will be de-identified prior to proceeding to the statistical analysis in accordance with the HIPAA Australia Privacy Act 1988 and the Australian Privacy Principles (APP) Guidelines, which provide the standard for de-identification of protected health information, the Australian De-Identification Decision-Making Framework, and AIHW guidelines for disclosure of secondary use health information and the Australian Privacy Act and Australian Privacy Principles [21–24].

**Minimum data requirements.** At a minimum, data collected for each participant should include:

- Identifiers (name, address, date of birth, contact details, comorbidities)

- Recruitment details (date of recruitment, date of consent, proof of consent, date that test packs were posted, symptomatology, duration of symptoms, presence of blood in faeces (if applicable))

- Sample collection details (where samples were collected (e.g., at home), who collected the sample (e.g., self-collect), date and time each sample was collected, date samples were submitted to pathology lab)

- Bulk faeces and rectal swab test results (date that lab testing was conducted for each sample, presence of visible faeces for the rectal swab only, positive/negative detection results for each sample)

- Follow-up information (date of follow-up attempts, which sample was collected first as reported by the participant, rough estimate of time difference between samples collected, adverse event details, preferred sample method)

**Deidentification methods.** De-identification involves two steps. The first is the removal of direct identifiers. The second is taking one or both of the following additional steps: 1) the removal or alteration of other information that could potentially be used to re-identify an individual, and/or 2) the use of controls and safeguards in the data access environment to prevent re-identification [25]. The study data will be de-identified using the following methods:

- Direct identifiers, such as names, addresses, will be removed as soon as the data integrity is verified and cleaned and prior to analysis, and never reported publicly.

- Indirect identifiers, such as dates of birth, zip codes, and medical codes, will be removed as soon as the data integrity is verified and cleaned and prior to analysis, and never reported publicly.

- Removal of the direct and indirect identifiers will be done as early as possible in the data cleaning phase to safeguard the data in case of re-identification.

- When presented or published, data will *always be aggregated and grouped* so that identification is not possible. We will suppress cells/rows or information with 5 or less persons and avoid reporting outliers in ranges (e.g., such as persons aged 107 years of age, or a death in a 2-month-old).

**Public release model.** The public release modelled data occurs after extensive cleaning and analysis. The released data is aggregated for descriptive analyses. The focus of the analysis is mathematical models that provide only group-wide estimates.

## Statistical methods

We will summarise safety and efficacy data using descriptive statistics, including means, medians, and frequencies. We will use statistical tests to compare groups (e.g., chi-square tests for categorical data, t-tests for continuous data). We will also adjust for multiple comparisons to control for type I error.

Logistic regression models will be used to assess accuracy in the presence of potential confounders. This includes sex (female/male), age (by age group), person collecting sample (self-collect/clinician collect), comorbidities, symptomatology (symptomatic/asymptomatic), duration of symptoms, and whether blood was present in the sample. Subgroup analysis will be conducted according to disease type (e.g., viral, or bacterial).

**Primary outcomes.** We will assess the sensitivity, specificity, positive and negative predictive value, and positive and negative agreement value, using 2x2 tables. The gold standard will be bulk faeces PCR testing. Sensitivity is the proportion of individuals with the disease who test positive with rectal swabs. We will assess the cycle threshold (CT) values using general linear models if the relationship is linear as expected and estimate the receiver operating curves and areas under the curve to assess for the potential to need differential CT value cutoffs. All results will be reported using 95% confidence intervals.

The R statistical suite will be used for all analyses and data visualisation. All estimates will be presented with 95% confidence intervals (CI). P-values <0.05 are deemed significant, and p-values <0.1 are considered trends.

**Secondary outcomes.** We will calculate the number and percentage of participants experiencing at least one adverse event, and the total number of adverse events reported. We will tabulate the distribution of adverse events by severity (mild, moderate, severe) for both the rectal swab and bulk faeces groups. Summarise AEs by expectedness (expected, unexpected) and relatedness (definitely, probably, possibly, unrelated) for each group. We will report the incidence rates of adverse events per 100 participant-days for each group, allowing for comparison over different follow-up periods. We will use chi-square tests or Fisher's exact tests to compare the incidence rates of adverse events between the rectal swab and bulk faeces groups. We will use ordinal logistic regression to analyse the severity of adverse events, treating severity as an ordinal outcome. We will include group (rectal swab vs. bulk faeces) as the main predictor and adjust for potential confounders (e.g., age, gender, mode of testing, and location of testing).

**Missing data.** To minimise occurrences of missing data, the data collection tool will be designed in a clear and easy-to-operate format. Any members of the research team who may be responsible for data collection and data input will be familiarised with data collection/storage tools and processes, which will ultimately reduce occurrences of missing data due to an avoidable error. Minimum data requirements are defined within the data collection platform to ensure instances of missing data are easily identifiable and can be followed up promptly (e.g., missing symptomatology details).

Data relevant to consent, participant identifiers and dual samples <u>must</u> be reported at the bare minimum. We will require each site to monitor and report missing data every 6 months. If a participant withdraws from the study after providing consent, the reason for withdrawal will be recorded and taken into account in data analysis and interpretation stages. Non-critical missing data will be managed via sensitivity analyses, and if amenable, multiple imputation will be used. In the event that a participant does not return the swab (critical data), that participant will be contacted to check if they wish to withdraw from the study. If the individual wishes to remain part of the study, they will be supported to provide the additional sample (e.g., required sample kit will be provided again if necessary). Participants will be encouraged to supply the bulk faeces sample at a minimum to complete routine clinical or public health care. If the participant withdraws from the study, or is unable to be contacted, the participant's data will be omitted.

**Interim analysis.** Interim analyses will be part of a yearly steering committee meeting or at every 100 participants recruited, whichever comes first. The trial will implement a modified alpha spending function to guide decisions on early stopping. Initially, at 100 participants, a non-inferiority margin of 0.35 will be applied. This threshold will decrease by 0.05 at each subsequent interim analysis, reaching a final margin of 0.10 after 500 participants have been recruited. If the observed difference exceeds the non-inferiority margin at an interim analysis, the trial will be stopped for inferiority.

**Objectives of interim analysis.**

- Recruitment Rates: Assess the rate of recruitment and discuss the barriers and facilitators to successful participant recruitment.

- Safety Monitoring: Ensure participant safety by identifying any significant adverse events early. Review adverse event reports, including severity, relatedness, and unexpected events. Analyse differences in safety profiles between the rectal swab and bulk faeces groups.

- Efficacy Assessment: Non-inferiority assessments will be performed to monitor for inferiority of rectal swab PCR in comparison with traditional bulk faeces testing. Evaluate the primary endpoint of difference of accuracy of specimen collection methods, the rate of pathogen detection.

- Trial Continuation Decision: Provide data to support decisions on continuing, modifying, or stopping the trial.

- Compliance and Protocol Adherence: Assess the adherence to study protocols, including the proper collection and handling of specimens.

**Oversight and monitoring**

The Coordinating Centre manages the operational and logistical aspects, ensuring smooth trial execution, and the overall trial implementation. Protocol evaluation and amendments will be discussed at regular Coordinating Centre trial conduct meetings.

The roles outlined here may be performed by more than one person. i.e., the Project Manager may also do the role of the Data Manager. The roles include:

- Principal Investigator (PI): Provides scientific leadership and oversees the entire trial. Ensures the trial adheres to regulatory requirements, supervises data analysis, and maintains overall responsibility for the study's conduct.

- Project Manager: Manages day-to-day operations of the trial. Coordinates activities across sites, manages timelines and budgets, and oversees staff at the coordinating center.

- Data Manager: Handles data collection, management, and analysis. Ensures data quality, integrity, and compliance with data protection regulations. Manages databases and performs statistical analyses.

- Clinical Research Coordinators: This role supports the PI and Project Manager in trial implementation. They are based on-site in a clinical setting and are responsible for the handling patient recruitment, informed consent, data entry, and follow-up activities.

- Regulatory Affairs Specialist: Ensures compliance with regulatory requirements. Manages submissions to ethics committees and regulatory bodies and maintains regulatory documentation.

- Quality Assurance (QA) Officer: Monitors trial conduct to ensure compliance with protocols and regulations. Conducts audits, manages deviations, and implements corrective actions.

- Statisticians: Provide statistical expertise for study design and analysis. Develop statistical analysis plans, perform interim and final analyses, and assist with interpretation of results.

- Administrative Staff: Provide administrative support to the coordinating center team. Handle documentation, scheduling, and communication tasks.

The Trial Steering Committee (TSC) provides independent oversight of the clinical trial. It ensures the trial is conducted according to the protocol and addresses any issues that arise during the study. The TSC provides independent oversight, ensuring scientific integrity, regulatory compliance, and participant safety. The TSC will have oversight of the interim analyses and key decisions such as a stop-early decision. Trial conduct will be discussed at TSC meetings and on demand by local ethics committees. The TSC will include:

- Independent Chairperson: Leads the TSC meetings and ensures unbiased oversight. Facilitates discussions, ensures all viewpoints are considered, and guides decision-making processes.

- Principal Investigator: Provides insights on trial progress and scientific aspects. Updates the TSC on trial status, protocol adherence, and emerging results.

- Independent Clinicians: Offer clinical expertise and independent perspectives. Review safety data, assess clinical implications, and provide recommendations based on their expertise.

- Regulatory Affairs Specialist: Ensures regulatory compliance and addresses regulatory issues. Provides updates on regulatory submissions, compliance, and any regulatory changes affecting the trial.

   **Decision rules.**

- Stopping for Safety: Predefined criteria for stopping the trial if a significant safety concern are identified; for instance, if the incidence of severe adverse events is above what is expected.

- Stopping for Efficacy: Criteria for stopping the trial early if rectal swabs show clear and overwhelming evidence of inferiority, based on predefined statistical thresholds and thus it is futile to continue. A cutoff of 0.35 difference in accuracy for inferiority with an alpha of 0.05 at interim analyses will be used to support decisions on continuing, modifying, or stopping the trial. As this is a diagnostic test trial, the decision to stop the trial must be multifactorial in nature and made by the Trial Steering Committee.

## Adverse event reporting

An adverse event (AE) is any untoward medical occurrence in a patient or clinical trial participant administered a medicinal product and that does not necessarily have a causal relationship with this treatment. An adverse reaction is any untoward and unintended response to an investigational medicinal product related to any dose administered. Probable and/or definitely related AEs are defined as adverse reactions for this trial.

A serious adverse event (SAE)/serious adverse reaction (SAR) is any adverse event/adverse reaction that results in death, is life-threatening, requires hospitalisation or prolongation of existing hospitalisation, results in persistent or significant disability or incapacity.

AEs will be classified according to:

1. Severity

   - Mild: Adverse events that are easily tolerated and do not interfere with daily activities. These could include minor discomfort or irritation at the site of the rectal swab.

   - Moderate: Events that cause some interference with daily activities but are not dangerous. Examples may include moderate pain, cramping, or a temporary increase in bowel movement frequency.

   - Severe: Events that significantly interfere with daily activities or pose a risk to health. Severe adverse events might involve intense pain, bleeding, or infection requiring medical intervention.

2. Expectedness

   - Expected Adverse Events: These are events that are known to occur based on previous studies or the nature of the procedures. For rectal swabs, this could include expected mild discomfort or minor bleeding.

   - Unexpected Adverse Events: Events that are not anticipated based on prior knowledge. Unexpected events could be severe allergic reactions, systemic infections, or unusual gastrointestinal symptoms.

3. Relatedness to Procedure

   - Definitely Related: There is a clear, direct link between the adverse event and the rectal swab procedure. An example is rectal bleeding immediately following the swab.

- Probably Related: There is a reasonable likelihood that the event is related to the procedure, but other factors could be involved. For instance, discomfort that occurs after the procedure and subsides soon afterward.

- Possibly Related: The event could be related to the procedure but could also be attributed to other causes. This might include abdominal pain occurring days after the procedure.

- Unrelated: The event has no apparent connection to the procedure. An example is an adverse event like a headache, which is likely due to other factors.

4. Type of Adverse Event

- Local Adverse Events: These are events occurring at the site of the rectal swab. They might include irritation, redness, or bleeding.

- Systemic Adverse Events: Events affecting the body beyond the local site, such as fever, nausea, or generalised infection.

- Procedure-Related Adverse Events: Specific to the process of specimen collection, like discomfort during the swab insertion or difficulties with the bulk faeces collection method.

- Non-Procedure-Related Adverse Events: Unrelated to the procedure itself, such as a common cold or unrelated illness occurring during the trial period.

## Dissemination plan

We will aim for a comprehensive local dissemination plan (including national conferences, local news networks, maximum reach for local population), progressing to international journals and conferences. Overall, this dissemination plan will reach consumers and the general public, clinicians, pathologists that may perform these tests and academics that may use this in their future work.

## Discussion

We will recruit participants through a streamlined referral process involving clinicians and local study coordinators. Initially, clinicians who encounter potential participants meeting the study criteria during routine medical visits will introduce the study to these patients. They will provide a brief overview of the study's purpose, procedures, and potential benefits, and assess the patients' interest in participating. If the patient expresses interest, the clinician will refer them to the study coordinator.

The study coordinator will then be responsible for the enrollment process. This will begin with contacting the referred patients to discuss the study in greater detail, answer any questions they may have, and obtain informed consent. Once consent is obtained, the coordinator will enroll the participant in the study and provide them with a rectal swab and bulk faeces testing kit. The coordinator will also give clear instructions on how to use the kits, ensuring the participant understands the procedures for collecting and sending samples. Throughout the study, the coordinator will remain available to support the participants, addressing any concerns and ensuring compliance with the study protocols. This systematic approach ensures thorough participant engagement and adherence to the study requirements. The study team staff will be focused on removing barriers for the participants. Firstly, making it easy for the swabs and faeces specimen containers to get to the participant (via overnight mail), and then ensuring ease of delivery of the specimens to the pathology laboratories or collection points.

The bulk faeces specimen and the rectal swab will be transported at 4°C and be retained for minimum one month as per standard laboratory protocols for such materials. A small sample (aliquot) of the bulk faeces specimen and the rectal swab will be retained for minimum one year after submission, or until the end of the project (whichever is later).

The bulk faeces test will be used to report positive or negative test results. For samples that are discordant (e.g., swab negative but bulk faeces positive for organism, or vice versa), results should be confirmed through additional testing according to internal laboratory protocol.

The trial process has been refined through consultation with the local Consumer Research Engagement Group (CREG). Continued engagement with the CREG throughout the lead-up to recruitment will help ensure that the study remains relevant, acceptable, and responsive to the needs and perspectives of the broader community.

## Supporting information

**S1 File. Completed SPIRIT checklist.**
(PDF)

**S2 File. Protocol approved by ethics committee.**
(DOCX)

## Acknowledgments

All authors would like to acknowledge the staff at participating clinics/recruitment sites, as well as Pathology Queensland staff.

## Author contributions

**Conceptualization:** Nicolas Roydon Smoll.

**Formal analysis:** Michael Thomas, Nicolas Roydon Smoll.

**Funding acquisition:** Madisen Sadie Roser, Nicolas Roydon Smoll.

**Investigation:** Megan Young, Shradha Subedi, Donna Barnekow, Alyssa Pyke, Jacob Tickner, Michael Thomas, Sanmarie Schlebusch.

**Methodology:** Megan Young, Dustylee Williams, Shradha Subedi, Donna Barnekow, Alyssa Pyke, Jacob Tickner, Glen Hewitson, Michael Thomas, Gulam Khandaker, Sanmarie Schlebusch, Nicolas Roydon Smoll.

**Project administration:** Madisen Sadie Roser, Megan Young, Nicolas Roydon Smoll.

**Resources:** Megan Young, Dustylee Williams, Glen Hewitson, Michael Thomas, Sanmarie Schlebusch, Nicolas Roydon Smoll.

**Software:** Nicolas Roydon Smoll.

**Supervision:** Megan Young, Michael Thomas, Sanmarie Schlebusch, Nicolas Roydon Smoll.

**Validation:** Nicolas Roydon Smoll.

**Visualization:** Nicolas Roydon Smoll.

**Writing – original draft:** Madisen Sadie Roser, Megan Young, Dustylee Williams, Shradha Subedi, Donna Barnekow, Alyssa Pyke, Jacob Tickner, Gulam Khandaker, Sanmarie Schlebusch, Nicolas Roydon Smoll.

**Writing – review & editing:** Madisen Sadie Roser, Megan Young, Dustylee Williams, Shradha Subedi, Donna Barnekow, Alyssa Pyke, Jacob Tickner, Glen Hewitson, Michael Thomas, Gulam Khandaker, Sanmarie Schlebusch, Nicolas Roydon Smoll.

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
