## [Decision Letter · Decision Letter 0]

5 May 2025

Dear Dr. Smoll,

Thank you for submitting your manuscript to PLOS ONE. After careful consideration, we feel that it has merit but does not fully meet PLOS ONE’s publication criteria as it currently stands. Therefore, we invite you to submit a revised version of the manuscript that addresses the points raised during the review process.

**ACADEMIC EDITOR:**

Please modify the comments given mainly from reviewer one which is a major revieion is requested.

We look forward to receiving your revised manuscript.

Kind regards,

Tsegaye Alemayehu, Msc

Academic Editor

PLOS ONE

Reviewers' comments:

Reviewer's Responses to Questions

**Comments to the Author**

1. Does the manuscript provide a valid rationale for the proposed study, with clearly identified and justified research questions?

Reviewer #1: Yes

Reviewer #2: Yes

Reviewer #3: Yes

2. Is the protocol technically sound and planned in a manner that will lead to a meaningful outcome and allow testing the stated hypotheses?

Reviewer #1: Partly

Reviewer #2: Yes

Reviewer #3: Yes

3. Is the methodology feasible and described in sufficient detail to allow the work to be replicable?

Reviewer #1: No

Reviewer #2: Yes

Reviewer #3: Yes

4. Have the authors described where all data underlying the findings will be made available when the study is complete?

Reviewer #1: No

Reviewer #2: Yes

Reviewer #3: Yes

5. Is the manuscript presented in an intelligible fashion and written in standard English?

Reviewer #1: Yes

Reviewer #2: Yes

Reviewer #3: Yes

You may also provide optional suggestions and comments to authors that they might find helpful in planning their study.

Reviewer #1: This study protocol describes a single-group diagnostic accuracy comparative trial evaluating the effectiveness of rectal swabs as an alternative to bulk feces for PCR testing in diagnosing enteric infections. The protocol plans to compare the diagnostic performance (sensitivity, specificity, PPV, and NPV) of rectal swabs against bulk feces as the gold standard in various clinical and community settings. If validated, this approach could improve patient compliance, reduce healthcare costs, and streamline the diagnostic process for enteric pathogens. This work addresses a significant public health and clinical problem by potentially improving enteric pathogen diagnosis. The study includes paired sampling to allow direct comparison of diagnostic accuracy, and comprehensive Outcome Measures assessing diagnostic performance, patient acceptability, and safety outcomes. However, I have the following concerns and recommendations:

1. The sample size calculation is based on sensitivity estimates but does not account for variability across different enteric pathogens. I suggest justifying the assumed disease prevalence and sensitivity estimates. Consider a stratified analysis or power calculation for key pathogen subgroups.

2. The study assumes that bulk feces PCR testing is a definitive gold standard, which may not always be the case. I suggest acknowledging potential limitations of bulk feces PCR testing and consider adjudicating discordant results with additional confirmatory testing.

3. The protocol does not specify the handling of missing or unreturned rectal swabs or fecal samples. I would recommend that the authors outline a plan for handling missing data, including imputation strategies or sensitivity analyses.

4. The protocol mentions adjusted analyses for demographics but does not specify a formal multivariate model. Please consider using logistic regression models or generalized estimating equations to adjust for potential confounders.

5. The interim analysis plan lacks clear predefined statistical thresholds for stopping due to futility or superiority. I would recommend defining explicit stopping rules for early termination based on interim sensitivity/specificity analyses.

6. The study population is not clearly defined in terms of demographics and potential selection biases. I suggest discussing how the study findings will generalize to broader populations, including pediatric patients and immuno-compromised individuals.

7. The authors state that individual-level data will not be publicly available. Could authors provide a data-sharing plan for de-identified aggregated results following FAIR principles?

Reviewer #2: The planned protocol is written in adequate detail and in clear language, and is feasible. The conflicting results described in literature make this a worthwhile study

Reviewer #3: Comments

1. The authors should include detail regarding the duration of diarrhea symptom before participants' presentation to the general practitioner, as this may influence viral detection outcomes. This should be incorporated into the study’s inclusion and exclusion criteria.

2. The authors indicate that hepatitis A or E will be identified through laboratory testing; therefore, it would be helpful to clarify whether the study will include participants with bloody or non-bloody diarrhea. This distinction should be addressed within the study’s inclusion and exclusion criteria.

**Do you want your identity to be public for this peer review?** For information about this choice, including consent withdrawal, please see our Privacy Policy

Reviewer #1: No

Reviewer #2: No

Reviewer #3: No

---

## [Author Response · Author response to Decision Letter 1]

17 Jun 2025

Dear Editor,

Thank you for your consideration of this manuscript. We appreciate the time and feedback that each Reviewer has provided, and we gratefully welcome any further feedback that may result from submission of the revised manuscript.

We have carefully considered each Reviewer’s comments. Please find below a response to each point of feedback which required action:

Reviewer #1:

1. The sample size calculation is based on sensitivity estimates but does not account for variability across different enteric pathogens. I suggest justifying the assumed disease prevalence and sensitivity estimates. Consider a stratified analysis or power calculation for key pathogen subgroups.

Thank you for pointing this out. We have included the following passage to explain this further:

“These estimates are based on general sensitivity assumptions and a simplified disease prevalence model, with the literature reporting non-specific prevalence to be approximately 25%, and sensitivity estimates typically >90%.[17-20] These estimates may change slightly as recruitment progresses and the distribution of pathogens becomes clearer. To address potential variability across different pathogens, stratified analyses will be conducted. Power calculations for key pathogen subgroups indicate that, for a sensitivity of 95% (with a range of 90-99%), a minimum of 73 positive samples is required per pathogen subgroup.”

2. The study assumes that bulk feces PCR testing is a definitive gold standard, which may not always be the case. I suggest acknowledging potential limitations of bulk feces PCR testing and consider adjudicating discordant results with additional confirmatory testing.

We agree that further testing should be conducted to confirm mismatched results. We have included the following in the manuscript:

“The bulk faeces sample is used as the control sample in this study as this sample type is currently used in routine clinical practice. In this context, bulk faeces PCR testing is considered the gold standard. It is acknowledged that bulk faces PCR testing may be associated with technical or interoperability limitations. To minimise the uncertainty that this could pose on testing results, paired samples that do not produce the same results will be managed internally according to laboratory protocol.”

3. The protocol does not specify the handling of missing or unreturned rectal swabs or fecal samples. I would recommend that the authors outline a plan for handling missing data, including imputation strategies or sensitivity analyses.

Thank you for pointing this out as it is a highly critical aspect of the trial. Given that it is a minimum requirement for both samples to be submitted, we have included the following passage in the manuscript:

“Non-critical missing data will be managed via sensitivity analyses, and if amenable, multiple imputation will be used. In the event that a participant does not return the swab (critical data), that participant will be contacted to check if they wish to withdraw from the study. If the individual wishes to remain part of the study, they will be supported to provide the additional sample (e.g. required sample kit will be provided again if necessary). Participants will be encouraged to supply the bulk faeces sample at a minimum to complete routine clinical care. If the participant withdraws from the study, or is unable to be contacted, the participant’s data will be omitted.”

4. The protocol mentions adjusted analyses for demographics but does not specify a formal multivariate model. Please consider using logistic regression models or generalized estimating equations to adjust for potential confounders.

Thank you for this suggestion, we have included the following sentence to clarify the type of modelling that will be used:

“Logistic regression models will be used to assess accuracy in the presence of potential confounders. This includes sex (female/male), age (by age group), person collecting sample (self-collect/clinician collect), comorbidities, symptomatology (symptomatic/asymptomatic), duration of symptoms, and whether blood was present in the sample.”

5. The interim analysis plan lacks clear predefined statistical thresholds for stopping due to futility or superiority. I would recommend defining explicit stopping rules for early termination based on interim sensitivity/specificity analyses.

We agree, the rules for stopping the study were not well described. We have reorganised the text to clarify:

“The trial will implement a modified alpha spending function to guide decisions on early stopping. Initially, at 100 participants, a non-inferiority margin of 0.35 will be applied. This threshold will decrease by 0.05 at each subsequent interim analysis, reaching a final margin of 0.10 after 500 participants have been recruited. If the observed difference exceeds the non-inferiority margin at an interim analysis, the trial will be stopped for inferiority.”

6. The study population is not clearly defined in terms of demographics and potential selection biases. I suggest discussing how the study findings will generalize to broader populations, including pediatric patients and immuno-compromised individuals.

Thank you for raising this point. We have intentionally left the inclusion criteria broad to ensure the trial can reflect clinical practice. We can certainly understand how this was poorly communicated. We have included the following passage to address this:

“The generalisability of this trial is supported by its relatively broad inclusion and exclusion criteria, which are designed to reflect routine clinical practice. Because the trial aims to evaluate the use of rectal swabs as an alternative to bulk faeces sampling, eligibility is not restricted (e.g. based on symptom presentation), allowing for the inclusion of participants with diverse clinical profiles. This inclusive approach will mimic routine clinical practice, and facilitate recruitment across a broad range of demographic, clinical, and socioeconomic backgrounds, enhancing the external validity of the findings. Vulnerable groups, such as comorbid or immunocompromised individuals, are identified in the recruitment phase and this data is recorded to be used in the analyses. The lower age limit of 18 years has been set due to ethical and regulatory considerations in recruiting children and performing rectal swabs on a vulnerable group. While children are not included in this study, the trial findings relevant to rectal swab accuracy are expected to be applicable to all ages.”

7. The authors state that individual-level data will not be publicly available. Could authors provide a data-sharing plan for de-identified aggregated results following FAIR principles?

Due to ethics constraints, we are unable to share data or store data externally to Queensland Health. However, we will provide as much relevant information as possible included in future RecSwabFaeces publications. We absolutely understand the concern with adhering to FAIR principles, and we have provided greater detail to address this:

“In accordance with ethics approval and participant confidentiality requirements, the full individual-level dataset cannot be publicly shared. However, all deidentified and aggregated data relevant to the reported findings will be made available through future RecSwabFaeces publications. Metadata and study documentation (e.g., protocols, workplace instructions, data dictionary) will be shared in publications, likely as support materials, to increase transparency, while remaining within the limits of ethical and legal constraints. The scalable design of the trial supports the reproducibility of the trial.”

Reviewer #3:

1. The authors should include detail regarding the duration of diarrhea symptom before participants' presentation to the general practitioner, as this may influence viral detection outcomes. This should be incorporated into the study’s inclusion and exclusion criteria.

2. The authors indicate that hepatitis A or E will be identified through laboratory testing; therefore, it would be helpful to clarify whether the study will include participants with bloody or non-bloody diarrhea. This distinction should be addressed within the study’s inclusion and exclusion criteria.

Thank you for highlighting this, these are great points. We have decided not to include duration of symptoms (such as diarrhoea), or presence/absence of blood in the inclusion/exclusion criteria, rather we will capture this information as variables to be included in the final multivariate model. Reporting these findings will be critical. We have not included these factors in our inclusion/exclusion criteria is because we aim to approximate routine clinical care which will see that a wide range of clinical circumstances require testing, and we are careful not to restrict the potential findings of the trial. Further, our inclusion criteria do not require that a person be ill to be recruited as we are also testing close contacts. We have included these variables as data requirements and the following sentences in our manuscript to address this:

“Logistic regression models will be used to assess accuracy in the presence of potential confounders. This includes sex (female/male), age (by age group), person collecting sample (self-collect/clinician collect), comorbidities, symptomatology (symptomatic/asymptomatic), duration of symptoms, and whether blood was present in the sample.”

On behalf of all authors, I would like to thank the Reviewers for their invaluable feedback and support in improving the quality and clarity of our protocol. We look forward to receiving the outcome of the revised submission.

Sincerely,

Dr. Nicolas Smoll

---

## [Decision Letter · Decision Letter 1]

20 Jul 2025

Rectal Swabs vs Bulk Faeces PCR testing for the diagnosis of enteric conditions (RecSwabFaeces): Protocol for a single group diagnostic accuracy comparative trial

PONE-D-25-10097R1

Dear Dr. Nicolas Roydon Smoll,

We’re pleased to inform you that your manuscript has been judged scientifically suitable for publication and will be formally accepted for publication once it meets all outstanding technical requirements.

Kind regards,

Tsegaye Alemayehu, Msc

Academic Editor

PLOS ONE

Additional Editor Comments (optional):

Reviewers' comments:

Reviewer's Responses to Questions

**Comments to the Author**

1. Does the manuscript provide a valid rationale for the proposed study, with clearly identified and justified research questions?

Reviewer #1: Yes

2. Is the protocol technically sound and planned in a manner that will lead to a meaningful outcome and allow testing the stated hypotheses?

Reviewer #1: Yes

3. Is the methodology feasible and described in sufficient detail to allow the work to be replicable?

Reviewer #1: Yes

4. Have the authors described where all data underlying the findings will be made available when the study is complete?

Reviewer #1: Yes

5. Is the manuscript presented in an intelligible fashion and written in standard English?

Reviewer #1: Yes

You may also provide optional suggestions and comments to authors that they might find helpful in planning their study.

Reviewer #1: Overall, the authors have thoroughly and effectively responded to each of my comments, and the revised manuscript now robustly addresses the methodological and statistical issues previously identified.

**Do you want your identity to be public for this peer review?** For information about this choice, including consent withdrawal, please see our Privacy Policy

Reviewer #1: No

---

## [Editor Report · Acceptance letter]

PONE-D-25-10097R1

PLOS ONE

Dear Dr. Smoll,

I'm pleased to inform you that your manuscript has been deemed suitable for publication in PLOS ONE. Congratulations! Your manuscript is now being handed over to our production team.

Kind regards,

on behalf of

Dr. Tsegaye Alemayehu

Academic Editor

PLOS ONE